# Recent Advance of Intelligent Packaging Aided by Artificial Intelligence for Monitoring Food Freshness

**DOI:** 10.3390/foods12152976

**Published:** 2023-08-07

**Authors:** Xiaoxuan Li, Danfei Liu, Yumei Pu, Yunfei Zhong

**Affiliations:** School of Packaging and Materials Engineering, Hunan University of Technology, Zhuzhou 412007, China; m22077300009@stu.hut.edu.cn (X.L.); m18080502003@stu.hut.edu.cn (D.L.); m21085600005@stu.hut.edu.cn (Y.P.)

**Keywords:** intelligent packaging technology, artificial intelligence, food freshness detection, deep learning algorithms, food industry

## Abstract

Food safety is a pressing concern for human society, as it directly impacts people’s lives, while food freshness serves as one of the most crucial indicators in ensuring food safety. There exist diverse techniques for monitoring food freshness, among which intelligent packaging based on artificial intelligence technology boasts the advantages of low cost, high efficiency, fast speed and wide applicability; however, it is currently underutilized. By analyzing the current research status of intelligent packaging both domestically and internationally, this paper provides a clear classification of intelligent packaging technology. Additionally, it outlines the advantages and disadvantages of using intelligent packaging technology for food freshness detection methods, while summarizing the latest research progress in applying artificial intelligence-based technologies to food freshness detection through intelligent packaging. Finally, the author points out the limitations of the current research, and anticipates future developments in artificial intelligence technology for assisting freshness detection in intelligent packaging. This will provide valuable insights for the future development of intelligent packaging in the field of food freshness detection.

## 1. Introduction

Food safety, nutrition and food security are interconnected. According to the World Health Organization (WHO), approximately 10% of the global population (60 billion individuals) suffer from foodborne illnesses annually, resulting in 4.2 million deaths and a loss of 33 million disability-adjusted life years (DALYs) [1]. Food-borne illnesses can be prevented [2]. The WHO plays a global leadership role in investing and coordinating action across multiple sectors to establish strong and resilient national food safety systems, while also providing consumers with the tools necessary to choose safe food. As a result, people’s attention to food safety issues is increasing, and the freshness of food is one of the most important factors that affect food safety.

Packaging is the most efficient means of ensuring food safety, preserving food quality, and mitigating food waste. Traditional packaging provides a more passive protection for products by utilizing the barrier properties of the packaging material [3,4]. The active packaging technology enhances the protective properties of packaging by incorporating antibacterial agents, antioxidants, releasing agents and absorbents to actively regulate its internal environment [5,6,7]. However, the absence of real-time monitoring for food freshness in active packaging poses a potential threat to consumer health. Consequently, novel technologies and methods are being explored to enhance food freshness. 

Intelligent packaging, as an emerging method of packaging [8], offers real-time product tracking [9], convenient information exchange, rapid detection of food freshness and wide applicability [10], convenient information exchange, rapid detection of food freshness [11], and wide applicability [12]. It can enhance the safety and healthiness of foods. Precise detection of food freshness can be achieved through advanced techniques such as pattern recognition and deep learning.

Therefore, the purpose of this review is to provide a concise and comprehensive summary of the latest advancements in artificial intelligence technology for intelligent packaging, with a focus on freshness detection. Therefore, this paper provides a brief overview of the current research status on intelligent packaging both domestically and internationally. It clarifies the classification of intelligent packaging technologies while outlining their advantages and disadvantages in food freshness detection methods. Additionally, it summarizes the progress made in applying artificial intelligence technologies to detect food freshness through intelligent packaging. Finally, the author highlights the limitations of current research and anticipates future directions for artificial intelligence technology in assisting with freshness detection in intelligent packaging. This will provide valuable insights for the future development of intelligent packaging in the field of food freshness detection.

## 2. Intelligent Packaging Technology Overview

In 2005, Yam et al. [13] proposed to define intelligent packaging as a packaging system with intelligent functions such as detection, perception, recording, tracking, communication, and logic (Figure 1). This system can play a good role in extending shelf life [14], enhancing product safety, improving product quality, providing effective information, and warning product problems [15]. Among many definitions of intelligent packaging [16], this definition is relatively comprehensive and extensive, and is currently used by most scholars and institutions [17].

At present there are a number of intelligent packaging products; for example, the University of California, Berkeley and Stanford University jointly developed intelligent packaging bags; Germany Kuhne developed a company that can be based on different external conditions and change the breathing strength, pH, gas composition and other parameters to achieve real-time detection and control of different food freshness of intelligent packaging bags; a Dutch company developed self-regulating intelligent packaging bags according to changes in external environmental conditions; Zhejiang University in Hangzhou, China, also developed self-regulating intelligent packaging bags according to changes in different external conditions. These products have to some extent improved current food freshness testing technology, which has limitations, high costs, slow speed and other shortcomings, but there are still many shortcomings to be further improved and perfected.

### 2.1. Intelligent Packaging Technology Definition

Intelligent packaging technology is of great significance in product protection, transportation, storage, sales and consumer experience. It can not only increase the value of the product, but also extend the shelf life of the product, resulting in better economic benefits. With the continuous development of socioeconomic levels and scientific technology, intelligent packaging technology is being more widely used in the market. This type of technology is a new packaging model that combines modern packaging materials, electronic information, sensor technology and high-tech technology such as the Internet. It utilizes modern electronic information technology to control and manage packaged products in order to extend their shelf life and improve their quality and quantity. In addition, intelligent packaging technology opens up new business opportunities for new digital businesses, making it suitable for more areas of Industry 4.0 [18].

### 2.2. The Link between Intelligent Packaging Technology and Intelligent Packaging

Intelligent packaging technology is a multidisciplinary crossover, multidisciplinary integration and multilevel in-depth research system. It is composed of material technology, information technology, microelectronics technology, automatic control technology, communication technology, etc. It involves many fields such as material synthesis, information storage, wireless communication, information identification and processing. It is a new knowledge system based on matter, with information as the core, materials as the carrier and technology as the means to study the laws of interaction between matter and information and their applications.

The main function of intelligent packaging is to use packaging materials to achieve protection and control of products, including monitoring and regulation of product quality and safety; identification and anti-counterfeiting of products; safety protection and tracking of products; and automation of production processes. Intelligent packaging technology is the application of intelligent packaging in production practice. It is the main component of intelligent packaging, which aims to improve the quality of products and increase their safety and reliability. In this sense, intelligent packaging is in fact a quality control system. The relationship between intelligent packaging technology and intelligent products is complementary [13]. Any intelligent packaging product is formed based on the corresponding intelligent technology, and it must be supported by the corresponding intelligent technology.

## 3. Classification and Application of Intelligent Packaging Technology

There are two main methods of classifying intelligent packaging technologies in China: (1) distinguishing by indicator class, sensor class and information technology class [19,20,21]; (2) distinguishing by functional material class, functional structure class and information technology class [22]. These two classifications have respectively summarized intelligent packaging technologies from multiple aspects, but there is a more vague and ambiguous definition between the categories; and with the continuous emergence of new technologies and their application in intelligent packaging [23,24], the drawbacks of this classification method have been further amplified. In order to avoid category confusion, this paper takes the definition of intelligent packaging as a starting point and combines it with the emerging intelligent packaging technologies to classify intelligent packaging technologies into two types according to the direct factor detection category and the indirect factor detection category.

### 3.1. Direct Factor Detection Category

#### 3.1.1. Freshness Monitoring Technology

As people’s living standards continue to improve, the evaluation of food quality is no longer based solely on the degree of spoilage, but the evaluation of food freshness is an important direction for food quality evaluation in the future. The freshness of food is related to its own metabolism and is influenced by the type of food, microbial species, storage conditions and packaging methods. Freshness determination techniques are usually used to predict the freshness of food by the relationship between the metabolites of the food (e.g., glucose, organic acids, ethanol, volatile nitrogen compounds, biogenic amines, carbon dioxide, ATP degradation products, and sulfuric compounds) [25,26] and the indicators. To be able to be in contact with the compounds, the freshness indicators must be placed inside the packaging. Depending on the indicator, this information can be detected by different methods.

Oxygen and carbon dioxide are considered the most important gases because they have a global impact on metabolism, which has a huge impact on the quality of packaged fruits and vegetables. These two gases are important parts of the respiration and photosynthesis process of vegetables and fruits. After cutting fruits and vegetables, photosynthesis stops while breathing continues. In respiration, oxygen is consumed through a chemical reaction that provides energy for vegetables, emitting carbon dioxide as a result of this reaction. The proportion of these gases inside a package affects the freshness, hardness and color of the packaged product [27]. Therefore, keeping the combination of these gases within a certain range within the package, the so-called modified atmosphere packaging (MAP), is essential to extend the storage time and shelf life of packaged fruits and vegetables.

In food animals (including aquatic animals, livestock animals, and meat lumps and meat pieces that are portions thereof), the metabolic mechanism known as the ATP (adenosine triphosphate) cycle works while the food animals are alive, thus keeping decay from progressing. However, after death, the cycle no longer works, and ATP decomposes to ADP (adenosine diphosphate), and further decomposes to AMP (adenosine monophosphate), IMP (inosinic acid), AdR (adenosine), HxR (inosine) and Hx (hypoxanthine). Furthermore, although an increase in the amount of IMP improves flavor by creating umami, the amount of IMP generated in aquatic animals is less than that in livestock animals, and at the same time that IMP is being generated, decay rapidly progresses. For this reason, normally, when consumers purchase aquatic animals, they particularly use the “freshness”, as observed in the outer appearance, as an indicator thereof [28].

Generally, dairy product indicators concentrate on the detection of the first kind of alterations, which occur as a result of microbial growth (pH, gas composition, etc.). These changes are detected by the indicators and transformed into a response, usually a color change which can be easily measured and correlated with the freshness of food [29]. Depending on the food type, this information can be detected by different methods (see Table 1).

Different packaging has different characteristics; the choice of food packaging materials is not random and should first be combined with food characteristics, such as the form of food (solid, liquid, etc.), whether it has oxidation and volatility and whether it needs to be stored away from light. Secondly, we should consider the safety of food packaging materials, based on the perspective of food safety; we should understand the characteristics of the food itself and the protection conditions required. We should study the sensitive factors that affect the main components of food, especially the nutrients such as fat, protein and vitamins, including the influencing factors of light, oxygen, temperature, microorganisms, physics, and mechanics, in order to achieve its protective function and appropriately extend its storage life. We should study and master the packaging properties and applicable scope and conditions of packaging materials; master the relevant packaging technology methods, for a given food, in addition to the need to select the appropriate packaging and containers, and should also use the most appropriate packaging technology methods. The same kind of food can often use different packaging technology methods to achieve the same or similar packaging requirements and effects. For example, for foods that are easily oxidized, vacuum or inflatable packaging, or an enclosed deoxidizer, can be used for packaging. For food packaging, the change in climate conditions in the commodity circulation area is crucial, because the temperature has a great impact on the chemical change in the internal composition of the food, the food microorganism and the barrier property of the packaging material itself. We should master packaging testing methods. For example, for canned food with empty cans, we often need to determine the dissolution of its inner coating in the food, deoxidized packaging should determine the oxygen permeability of the packaging material, and moisture-proof packaging should determine the water vapor permeability of the packaging material.

The direction of development of intelligent packaging technology for freshness testing is becoming increasingly cost effective and non-destructive. Anusankari et al. [32] developed a new non-invasive, simple and inexpensive fluorescent probe that enables qualitative/quantitative analysis of changes in the concentration of metabolites such as oxygen and carbon monoxide in meat for the identification of meat freshness. Ezati et al. [33,34] developed alizarin freshness indicators based on alizarin dyes of cellulose, chitosan and amylocellulose using Total Volatile Basic Nitrogen (TVB-N) content as a test indicator and alizarin as a dye component, respectively. In the process of monitoring the freshness of ground beef and frozen rainbow trout fillets, the developed freshness indicators were found to have a good color change range in response to the TVB-N content.

The use of methods such as mobility spectrometry, multispectral imaging and gas chromatography allows for rapid and accurate analysis of metabolites in food, as well as monitoring the freshness of food. Cavanna et al. [35] used ion mobility spectrometry with gas chromatography to construct a data model that allows effective prediction of egg freshness, a method that has the advantages of fast detection, high sensitivity and efficiency. Cheng et al. [36] used multispectral imaging to monitor the freshness of fish fillets, using key bands in multispectral images to model the simultaneous prediction of TVB-N content, thiobarbituric acid reactant values and K values (freshness index) during the chemical deterioration of fish fillets, thus demonstrating the feasibility of developing a method to monitor the freshness of fish fillets in a rapid and real-time manner.

A piezoelectric sensor is a sensor that detects changes in the properties of a substance through changes in electrical potential. Ehrenberg et al. prepared a piezoelectric sensing material by using a polyvinyl alcohol (PVA) film as a substrate and coating a layer of graphene oxide (GO) on its surface, and used it as a signal transmitter for the freshness detection of pork. The results show that the method can achieve rapid detection of pork freshness and requires a small amount of sample with high accuracy in the prediction of pork freshness.

#### 3.1.2. Maturity Monitoring Technology

After being picked or processed, some foods enter storage, distribution as well as consumption, and at each stage, there is a risk of food maturation, which may have an impact on its freshness. Therefore, monitoring the degree of maturity of food can not only ensure the quality of food but also increase consumer confidence.

Diaz et al. [37] used gas chromatography tandem mass spectrometry to monitor volatiles in tequila at different maturation times to derive the relationship between volatile matter content and maturation time and to explore its utility in tequila quality control. Effective cumulative temperature is an important indicator to evaluate the maturity of vegetable and fruit crops. Lee et al. [38] developed a method to predict the maturity of melon, which monitors the effective cumulative temperature and predicts the maturity of melon with the help of time–temperature indicators (TTi) based on the Merad reaction.

### 3.2. Indirect Factor Detection Category

In addition to the food itself, environmental factors (temperature, humidity, mechanical stress, etc.) can also have an indirect effect on the safety and quality of food. Changes in the surrounding environment can lead to dynamic changes related to the quality of the food, which can accelerate spoilage. Therefore, in some intelligent packaging technologies, changes in environmental factors have been used as important indicators to evaluate the quality of food. The indirect factor detection category of intelligent packaging technologies refers to a class of sensors, indicators that monitor the external or internal environmental state of food packaging. 

Indirect factor detection intelligent packaging technologies mainly include TTi, leak indicators.

#### 3.2.1. Time–Temperature Indicators

Time–temperature indicator (TTi) is a new detection technology that is small, easy to operate and highly efficient, enabling effective monitoring of food quality and enhancing consumer confidence in food safety. TTi is a device or intelligent label that displays the cumulative time–temperature history of a product, which can use the time–temperature accumulation effect to monitor the product at all stages between manufacture, storage and transportation to the consumer It is widely used in the quality inspection of perishable products as it allows real-time monitoring of the temperature history of products at various stages between manufacturing, storage and transportation to the consumer, thus ensuring the actual quality and safety of food products [39].

TTi can be divided into four types, physical, chemical, biological and enzymatic, depending on how the color change in the indicator works. The physical type [40,41,42,43] TTi relies mainly on physical changes between colored substances (e.g., physical diffusion), which are all closely related to parameters such as temperature and time, and it contains diffusion type TTi, and nanotype TTi. The chemical type TTi [44,45,46,47] is used to change the color of molecules or compounds through a chemical reaction between them, and its reaction rate is influenced by temperature and time. The common chemical types, TTi, contain the polymeric type TTi, the photochromic type TTi, and the oxidation–reduction type TTi; the biological types [48,49,50,51] TTi are used to correct the color of the indicator by measuring the acid produced by microorganisms at a certain temperature, commonly used: lactic acid bacteria TTi, yeast TTi; in the enzymatic [52,53] TTi in which the hydrolysis and reaction of biological enzymes have a strong influence on the color change in TTi and correlate well with temperature. Compared to other TTi, enzymatic TTi has the advantages of stability, low cost and good controllability [54]. The two common types of enzymes used are hydrolysis reactions and enzyme catalysis.

#### 3.2.2. Leak Indicators

At every stage of the food supply chain, there is the potential for packaging leakage, which can lead to changes in the internal environment of the packaging and thus indirectly affect the quality of the food product. For example, the level of oxygen in a package can directly affect the rate of microbial growth and the degree of oxidation of oils and fats, which in turn affects the quality of the product. Leakage indicators or gas indicators alert consumers to product safety by monitoring changes in the gas content (e.g., CO, or O_2_) in food packaging. The low levels of CO_2_ in the air make it an ideal target for monitoring packaging leaks. Yusufu et al. [55] developed a colorimetric analysis-based air pressure indicator for vacuum packaging to indicate the integrity of the packaging by monitoring the level of CO_2_ in the vacuum packaging environment. The indicator relies on the color change in a pH chromogenic agent (o-cresolphthalein) dissolved in a non-aqueous solution of ethyl cellulose to respond to changes in CO_2_ content and can be used as an inexpensive indicator of food vacuum packaging integrity.

Colorimetric oxygen leakage indicator reagents are by far the most used. Most of these indicators use nanoscale particle photocatalytic materials to achieve controllability of redox dyes such as methylene blue. In tests, the indicator is usually sprayed on a plastic film to create an oxygen indicator film [56,57,58].

The direction of development for colorimetric oxygen leak indicators is towards lower costs and simpler production processes. Lawrie et al. [59] developed an inkjet-printable UV-activated oxygen indicator. The indicator consists of a redox dye (methylene blue), a semiconductor photocatalyst (TiO_2_) and a sacrificial electron donor (tartaric acid), which is sprayed onto a polyester film by inkjet printing to form an oxygen leak indicator film. Khankaew et al. [60] developed a novel UV-activated biotype oxygen leak indicator, which consists of polymer, biotype dye, nanosemiconductor (ZnO TiO_2_), electron donor (glycerol, sorbitol), and surfactant. Experimentally, it was demonstrated that there is a good correlation between the oxygen concentration and the color change in the indicator; meanwhile, its production process is simple and has a broad market application prospect.

However, colorimetric oxygen indicator reagents are expensive to use due to their high sensitivity to oxygen and the need to store them under anaerobic conditions. Wen et al. [61] developed a pressure-sensitive colorimetric oxygen indicator based on three components, methylene blue, glucose and NaOH, which can be used to effectively solve the challenge of visualizing high oxygen content by simply rupturing the chamber spacer membrane by pressure. Won et al. [62] proposed to use laccase, guaiacol and cysteine as raw materials for anaerobic storage of contrast oxygen indicators by modifying them in a double chamber and breaking the barrier layer by pressurization.

The redox dye used for the oxygen leak indicator has a leakage problem. Vu et al. [63,64,65] investigated three different oxygen indicators with different properties were prepared by coating different encapsulants on the indicator films, thus effectively solving the problem of dye leaching that occurs when dye-based oxygen indicator films come into contact with water, and also avoiding secondary contamination of food by the indicators. The leak indicator is suitable for monitoring the quality of meat products.

### 3.3. Information Aids

#### 3.3.1. Bar Code Technology

A barcode is a graphic identification symbol of a certain width and thickness that stores data and information by means of a special device, based on the basic principle of optical recognition. They can be divided into 1D and 2D barcodes according to their shape. Barcodes are inexpensive, easy to use and widely used to facilitate stock control, inventory recording and checkout [66]. Depending on the type, they have different storage capacities [67]. One-dimensional barcodes (Figure 2A) consist of black and white stripes of different widths, linearly arranged according to certain rules.

Two-dimensional barcodes (Figure 2B), on the other hand, are arranged in a two-dimensional geometry. Bar code technology enables information to be read quickly and without the need for additional equipment and personnel.

The Universal Product Code (UPC) barcode is mostly used now. Barcodes are used to track the location of every package along the supply chain as each one has a unique UPC. This type of data carriers can also be integrated with sensors and indicators to check the freshness of packaged vegetables through erasing or amending the barcode to make it unreadable.

There are many other examples that uses barcode labeled antibodies as biosensors. For example, a barcode capable of sensing the presence of pathogens and identifying whether a product is contaminated or not, known as the “Food Sentinel System”, is shown in Figure 3A. This system uses two barcodes, one for the information about the product which is readable by barcode reader, and another for contamination code symbol which is usually unreadable in case of safe food. The antibodies are labeled and made in the form of a membrane attached to a part of the barcode. When contamination is detected, one barcode will be unreadable and the other readable. This is because lines appear when antigens bind to the antibodies on the barcode or disappear due to their dissociation from the substrate in case of bacterial metabolite being present [68,69,70]. This system uses labeled antibodies on the membrane of the barcode to identify specific antigens. Another version of this technology is based on turning the membrane containing antibodies into red ink when a pathogen attaches to it, and so would not be readable by scanners. An illustration of this indicator is shown in Figure 3B [70,71]

A microbial TTI placed over barcode, known as TRACEO has the labelled barcode impregnated with lactic acid bacteria that changes the color of the barcode from transparent to opaque due to time expiration or indication of high temperature exposure of the package. Thus, this barcode will become unreadable showing that produce is no longer safe to consume [73].

#### 3.3.2. Radio Frequency Identification Technology (RFID)

RFID technology is a technology that uses wireless sensing for identification, which allows for contactless automatic identification of specific tags. RFID identification systems are generally composed of three parts: RFID tags, readers and data management systems, as shown in Figure 4 [74,75,76,77].

The RFID tag has a built-in antenna for signal transmission and a chip for data storage, a reader for collecting information inside the RFID tag, and a data management system for receiving and processing the information collected by the reader. According to the power supply method, RFID tags can be divided into active tags, passive tags and semi-active tags: active tags have built-in batteries and can interact with the reader for data; passive tags, also known as passive tags, have no internal power supply equipment and only activate the data transmission function when the reader/writer is close; semi-active tags also have internal power supply, but it has a big difference with active tags The semi-active tag also has an internal power supply, but it differs significantly from the active tag in that its battery is only used for the data storage function of the chip, and therefore it does not allow for real-time information interaction [78].

RFID technology is widely used in the meat, dairy, fisheries, bakery and beverage industries for its traceability, efficiency and non-contact benefits [79].

The combination of radio frequency identification technology and sensing technology will become the development trend for modern food quality and safety assurance. There are three important factors that affect the quality of food during storage and transportation: changes in acidity due to microbial growth, changes in humidity and changes in temperature [80]. Combining radio frequency identification technology with temperature sensing technology can effectively improve the management efficiency of the cold chain supply chain, reduce waste and lower costs. Shafiq [81] has developed a passive RFID temperature sensor that can be used to monitor the temperature of the cold chain environment for perishable food or pharmaceuticals, and it has designed a sensor that can be repurposed to control the cost of use. Lorite et al. [82] designed a critical temperature sensor based on the solvent phase change point to enable real-time monitoring of perishable foodstuffs using the principle of irreversible color change and in combination with RFID technology. Trebar et al. [83] argue that a combination of RFID technology and temperature sensors can be used to monitor the temperature changes in sea bass in logistics and determine the optimal preservation method. The invention can also be used to effectively monitor and analyze the environment of the inner bag to ensure food safety by using an infrared spectrometer in conjunction with a moisture sensor. Nair [84] developed a humidity sensor based on chipless RFID technology, which can monitor the moisture changes in the surrounding environment in real time and has good application prospects. Food quality is affected by many factors, and the combination of radio frequency identification (RFID) and multiparameter sensing technology can effectively achieve effective monitoring of food quality. Quintero [85] has developed a semi-active RFID intelligent tag with multiple parameter measurement capabilities to accurately measure room temperature, humidity and ammonia concentration.

At present, RFID technology is popular and in use, but there are still problems such as high costs, weakened signals or shielding. Modern printing technology to prepare chipless RFID tags can solve the cost problem of RFID. Feng Yi et al. [86] developed a chipless RFID tag that can monitor humidity, which can be printed directly on traditional packaging, greatly reducing the manufacturing cost of RFID tags. Shao Botao et al. [87] designed a chipless RFID tag based on the principle of capacitive/inductive resonance and combined it with an overprinting process to achieve a chipless RFID tag. The use of inkjet printing technology is an important method to reduce the manufacturing cost of RFID tags. WangYan [88] used inkjet printing technology combined with surface modification and chemical deposition processes to fabricate 1001 RFID flexible metal antennas, offering the possibility of low-cost RFID tagging.

#### 3.3.3. Augmented Reality (AR)

Augmented reality technology (AR) refers to the use of computer-created virtual images to augment the real environment, using a computer to simulate them, resulting in a virtual image that is superimposed on the real environment to create a hybrid reality and virtual environment, as shown in Figure 5 [89].

AR is composed of augmented reality display equipment and virtual scenes, and its steps can be summarized as scene capture, image recognition, image processing, and reality visualization, as shown in Figure 6. The camera of the helmet-mounted display records and collects the real environment through the user’s perspective, and transmits the recorded environmental data to the AR system. Then the AR system searches the collected environment, and when the set AR identification mark is found, the virtual image is accurately superimposed into the real environment to achieve the combination of virtual and reality.

Compared to RFID technology, AR technology has the advantage of triggering a new consumer experience and generating consumer interest. It is worth noting that AR technology does not require additional costs for consumers and retailers to purchase a device (which is possible through intelligent phones) during the experience segment, which certainly makes AR technology more competitive in the marketplace. Blippar has used AR technology in the design of children’s fudge packaging and embedded fun games into the AR system for product sales and promotions. Skywell Software has enhanced the interaction between merchandise and customers through AR technology and online content delivery with 3D tracking, enabling it to live stream games through intelligent packaging.

## 4. Artificial Intelligence Technology Used in Food Freshness Testing

Food freshness has a direct impact on its quality, and clever packaging can successfully extend its shelf life while minimizing financial losses brought on by food spoilage and deterioration. The creation of effective smart packaging should take precedence over everything else right now in order to provide a practical, quick, and efficient method of displaying changes in food quality in real time, as well as to lessen the health issues brought on by subpar food quality and safety [90]. However, the current research on the application of intelligent packaging technology in food freshness monitoring is still in its infancy, and the small number of sensors does not meet the demand for fast and accurate detection. At the same time, the sensors used for food freshness monitoring are expensive and sensitive to the environment, which seriously limits their application in intelligent packaging. In order to improve the intelligence and detection efficiency of intelligent packaging materials, in addition to developing new sensors with better performance, methods combining intelligent packaging materials with artificial intelligence technology have been introduced, and investment in research combining food freshness detection methods with artificial intelligence algorithms has also been strengthened.

Artificial intelligence technology is a theory based on computer technology, with data as the core, featuring expert systems and applying knowledge from mathematics and physics. Artificial intelligence is a multidisciplinary and marginal discipline, which consists of five main processes: perception, understanding, reasoning, learning and adaptation [91]. The core idea of artificial intelligence is to simulate human intelligence, and its development is closely related to factors such as computer hardware, software systems and learning algorithms. With the continuous development of electronic information technology, artificial intelligence technology will also be continuously improved and enhanced, and its application scope will be further expanded. The application of artificial intelligence in food freshness monitoring mainly includes data analysis and processing, pattern recognition and classification, and other applications of artificial intelligence in food freshness monitoring.

### 4.1. Deep Learning-Based Food Freshness Detection

A deep learning-based approach to food freshness detection has been proposed for the problem of odor quality assessment of meat, cereals, coffee, mushrooms, cheese, sugar, fish, beer and other beverages as well as food packaging materials. Two indicators, microorganisms on the food surface and volatile salt nitrogen (TVB-N), are mostly studied, and the variation patterns of the two indicators are analyzed by constructing a multilayer convolutional neural network model. Convolutional neural networks process food freshness data by constructing convolutional, pooling and fully connected layers between the input and output layers, among other operations [92].

In the convolutional layer, the input image is convolved point by point to output the feature vectors in the image; in the pooling layer, the feature vectors are multiscale normalized; in the fully connected layer, all vectors are weighted and summed to obtain the final output (Figure 7). The convolutional neural network can automatically extract features from the input image to achieve fast classification and recognition of food freshness data. The method can achieve rapid detection of food freshness and provide technical support for food quality and safety supervision.

For example, Palakodati et al. [93] proposed a model to avoid the spread of decay. The model classifies fresh and decaying fruits from the input fruit images. In this work, three types of fruits were used, such as apples, bananas and oranges. Convolutional neural networks (CNN) are used to extract features from the input fruit images and Softmax is used to classify the images into fresh and rotten fruits. The performance of the proposed model was evaluated on a dataset downloaded from Kaggle and produced an accuracy rate of 97.82%. The results show that the proposed CNN model is effective in classifying fresh and decaying fruits.

Kazi et al. [94] used various architectures of classical convolutional neural networks and residual convolutional neural networks in their study to classify three different types of fruits and their relative freshness determination. It was confirmed that the freshness of fruits could be determined more accurately using conventional convolutional neural networks and residual convolutional neural networks.

Guo et al. [95] combined a combined cross-reactive colorimetric barcode technique with a deep convolutional neural network (DCNN) to form a meat freshness monitoring system. The system is fast, accurate and non-destructive, enabling consumers and all stakeholders in the food supply chain.

Khaled [96] this developed an online classification system using deep convolutional neural networks (DCNN) for classifying bell peppers into five categories. According to export standards, crops are graded according to maturity stage and size, and then food grade bell peppers are retained, saving time while reducing manpower. Santana et al. [97] proposed to combine the random forest algorithm with artificially generated outliers in real samples for adulteration detection purposes, which can be used for freshness foreign matter detection.

### 4.2. Computer Vision-Based Food Freshness Detection

Computer vision technology is an emerging technology that is a multidisciplinary cross-fertilization of computer science, automatic control and image processing, and is an important part of the research field of artificial intelligence, with the ability to perceive, understand and manipulate information from the external world. As an emerging non-destructive testing technology [98], it has been widely used in food freshness testing with its fast and efficient features. This technique mainly uses image processing algorithms for image analysis, and combines pattern recognition and deep learning algorithms to process and analyze the image information to achieve food freshness detection.

Harnsoongnoen et al. [99] presented a non-destructive, non-invasive, low-cost, simple, real-time method for grading and monitoring the freshness of eggs based on density detection by machine vision and load cells. This method verifies that the freshness of eggs can be determined by density and has the potential to become an important measurement system for the poultry industry in the future. Kingshuk [100] used two different freshness assessment models, employing statistical methods such as principal component analysis (PCA) and supervised learning algorithms such as artificial neural networks (ANN) to investigate the different characteristics of mushroom images and classify them into fresh and spoiled categories. Observations show that the supervised learning models outperform the statistical models in terms of classification accuracy.

Image processing is a central part of computer vision-based detection of food freshness. Image processing usually includes image acquisition, image pre-processing and feature extraction. As different types of food require different image characteristics to be captured, the contrast between the target to be detected and the background, color characteristics and other information should be considered when image pre-processing is carried out. During image processing, image segmentation can be achieved according to differences in color, brightness and texture features between the detection target and the background, i.e., color quantization or brightness segmentation of food products. Feature extraction is the extraction and analysis of food freshness features, such as color, texture, shape and other information. As there is variability in the freshness index of different food products, different selection criteria need to be developed according to different types of food products when performing feature extraction (Figure 8).

Pattern recognition is the process of classifying multiple samples based on a specific feature. In food freshness detection, the segmented samples can be divided into two categories: “white zone” and “grey zone”. The “white zone” means that the segmented image is not uniformly colored and has more texture and detail; the “grey zone” means that the segmented image is uniformly colored and has less texture and detail. When the two categories of samples belong to two classes, pattern recognition can be used to extract classification features directly from the computer vision system to quickly determine the freshness of the food, mainly through principal component analysis, support vector machines and neural networks.

Abhishek et al. [101] were able to predict the suitability of mushrooms for consumption by classifying the freshness of mushroom samples, and this experiment also investigated the behavior of enzymatic browning of mushrooms. The classification accuracy of different classifiers was examined and it was concluded that the support vector machine (SVM) classifier had the highest classification accuracy of 80%.

This development by Khaled [100] of an online classification system using deep convolutional neural networks (DCNN) is also considered to be the latest technology in the field of machine vision-based classification. Bhargava et al. [102] dealt with various methods of pre-processing, segmentation, feature extraction and classification of fruit and vegetable quality based on color, texture, size, shape and defects, using computer vision to accomplish various classification and grading algorithms that can be used to detect the freshness of fruits and vegetables very well.

## 5. Conclusions and Future Perspectives

The application of artificial intelligence technology in food freshness testing has been widely adopted both domestically and internationally. Although some progress has been made in intelligent packaging materials, artificial intelligence algorithms, sensors and signal acquisition, there are still some shortcomings in practical applications, mainly in the following areas.

(1) There are few types of sensors for monitoring food freshness, and their performance is unstable. The type and performance of sensors have a significant impact on the effectiveness of food freshness detection; therefore, it is necessary to develop sensors that are suitable for different food freshness detection. Currently, the neural network algorithm, widely used in artificial intelligence algorithms, has strong recognition and learning abilities; however, its stability needs further improvement. With the continuous development and optimization of artificial intelligence technology, the combination of artificial intelligence algorithms and sensors has great potential for monitoring food freshness in the future. In addition, the performance of the sensor has a great impact on the detection effect of food freshness. For example, traditional electronic nose sensors are greatly affected by gas components in monitor freshness, but no new sensor has been developed specifically for these gas component characteristics.

(2) Artificial intelligence algorithms require a significant amount of time to compute and process vast amounts of data. The processing and analysis of a large amount of data in food freshness inspection necessitate numerous operations to acquire additional information, which can impose a significant burden on the food freshness inspection system. Contemporary artificial intelligence algorithms frequently leverage GPUs (Graphics Processing Units) to expedite data processing and analysis. However, GPUs are limited to processing 1D and 2D data only, which renders their acceleration performance inadequate for handling 3D and multidimensional data that are required in practical applications. To address this issue, it is possible to integrate multiple datasets in order to achieve diverse dimensions of data processing. Moreover, the stitching of multidimensional datasets necessitates the computation and manipulation of multiple temporal intervals, thereby resulting in a significant increase in computational time.

(3) Most food freshness detection algorithms utilize artificial neural networks, support vector machines, and other advanced techniques. For instance, the acquisition of satisfactory prediction performance by artificial neural networks necessitates a considerable amount of training samples; support vector machines’ classification accuracy is significantly influenced by sample size; and when applied to food freshness detection, support vector machines may generate substantial errors and their predictive precision cannot be guaranteed to reach 100%.

(4) The food freshness detection system exhibits slow information processing speed, prolonged response time, and extended signal acquisition time. At present, the application of artificial intelligence technology in food freshness detection mainly focuses on image processing and pattern recognition. For instance, to achieve better results in monitor food freshness, images need to be converted into chemical signals through recognition algorithms. Additionally, converting sensor response signals into chemical signals is also necessary for artificial intelligence algorithms used in sensor response signal processing. The artificial intelligence algorithm responsible for processing the sensor response signal must also convert it into a chemical signal and compare it with the food freshness result. Therefore, food freshness detection systems must enhance their information processing speed to meet the requirements of their applications. In addition, the process of freshness testing requires the collection of large amounts of raw data and their analysis, modeling, and prediction, which is time- and energy-consuming.

(5) The level of intelligence in artificial intelligence technology-assisted food freshness testing is low and lacks sustainable development capability. The current application of artificial intelligence technology in food freshness detection mostly remains in the experimental stage, and there are still many challenges to overcome for practical implementation of AI-assisted food freshness detection. Firstly, there is a lack of research on intelligent packaging materials, and no research results exist for intelligent packaging materials that can be used for AI-assisted food freshness testing. Secondly, there is a lack of comparative studies on the stability and reliability of different intelligent packaging materials under various storage conditions. Once again, there is a lack of analysis and discussion regarding the feasibility, influencing factors, advantages, and disadvantages of AI technology in the process of testing food freshness. Finally, there is a lack of discussion regarding the development of integrated AI technology-assisted systems for testing food freshness. As AI technology has obvious advantages over traditional chemical sensors, this paper will focus on the research progress of AI technology-assisted food freshness detection methods. However, there is little research on the combination of AI technology and chemical sensors for food freshness detection both domestically and internationally. Many studies are still in the experimental stage, and most AI-assisted food freshness detection methods involve data analysis and machine learning, which presents challenges when applying them to practical engineering.

In future research, AI technology can be further explored in the following areas: firstly, integrating AI technology with intelligent packaging materials to establish a correlation between intelligent packaging materials and food freshness through mathematical models; secondly, utilizing AI technology to evaluate food freshness and establishing an evaluation index system; thirdly, applying AI technology to the field of food freshness detection for identifying food freshness; fourthly, the integration of AI technology with conventional detection methods can enhance detection efficiency and facilitate prompt and precise assessment of food freshness.

## Figures and Tables

**Figure 1 foods-12-02976-f001:**
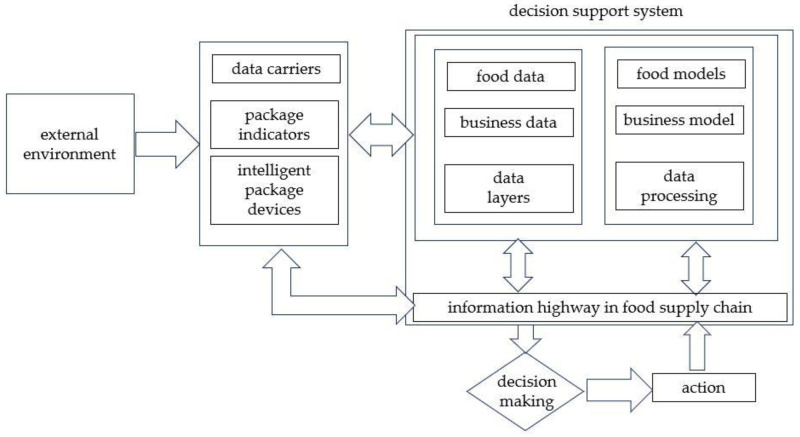
Intelligent packaging system in food supply chain [13].

**Figure 2 foods-12-02976-f002:**
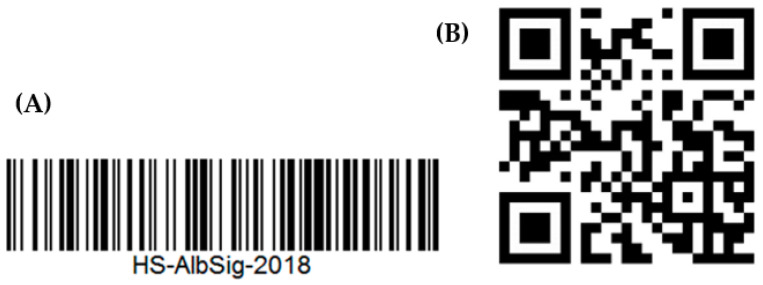
1D barcode (**A**); 2D barcode (**B**) [10].

**Figure 3 foods-12-02976-f003:**
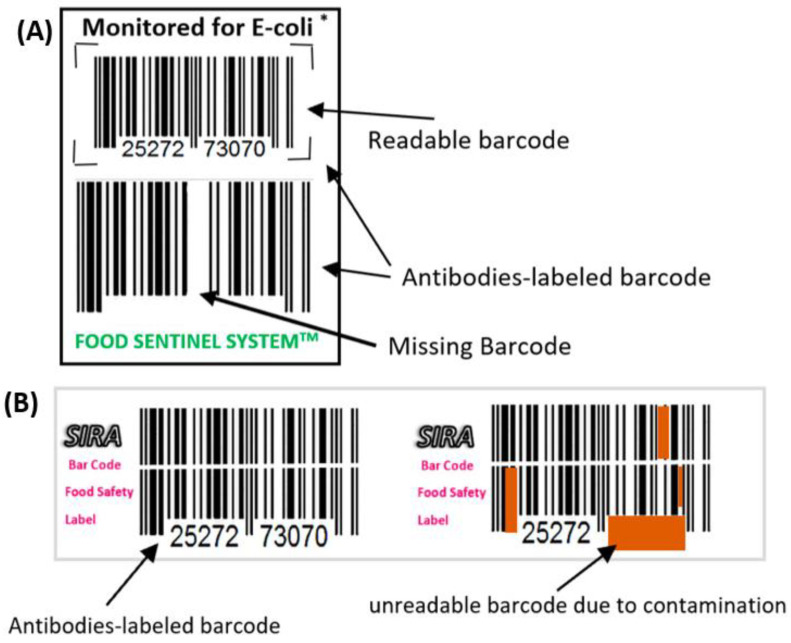
(**A**) Form of food sentinel system. (**B**) Another form of food sentinel system [72].

**Figure 4 foods-12-02976-f004:**
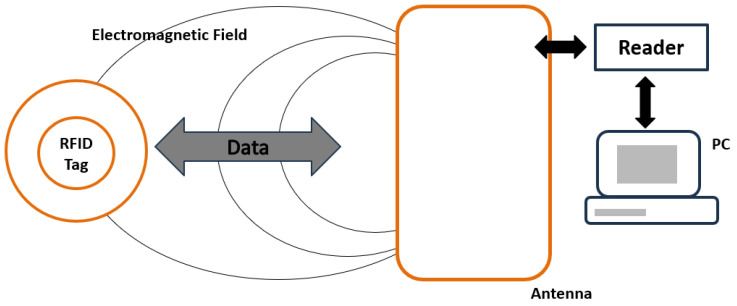
The working principle of radio frequency identification (RFID) tag, adapted from [10].

**Figure 5 foods-12-02976-f005:**
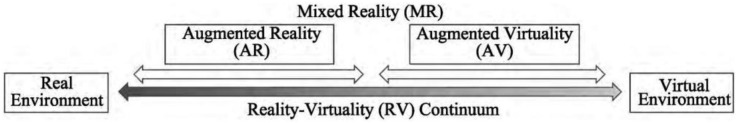
Reality-virtuality continuum [89].

**Figure 6 foods-12-02976-f006:**
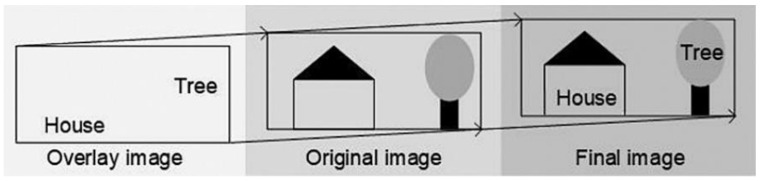
Working principle of augmented reality technology.

**Figure 7 foods-12-02976-f007:**
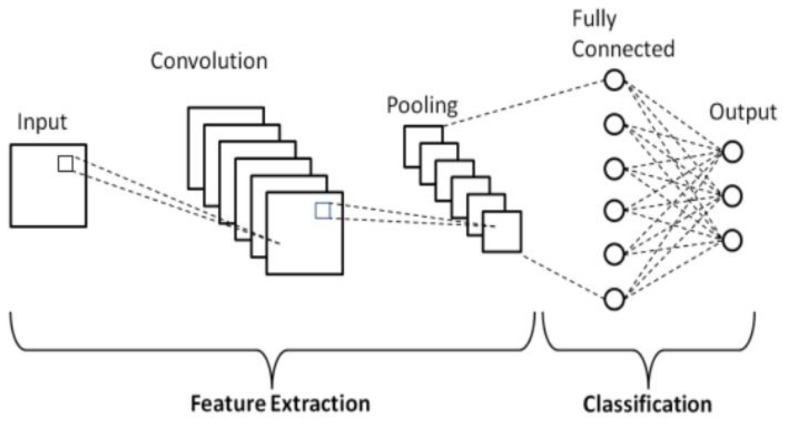
Basic CNN architecture for classification [93].

**Figure 8 foods-12-02976-f008:**
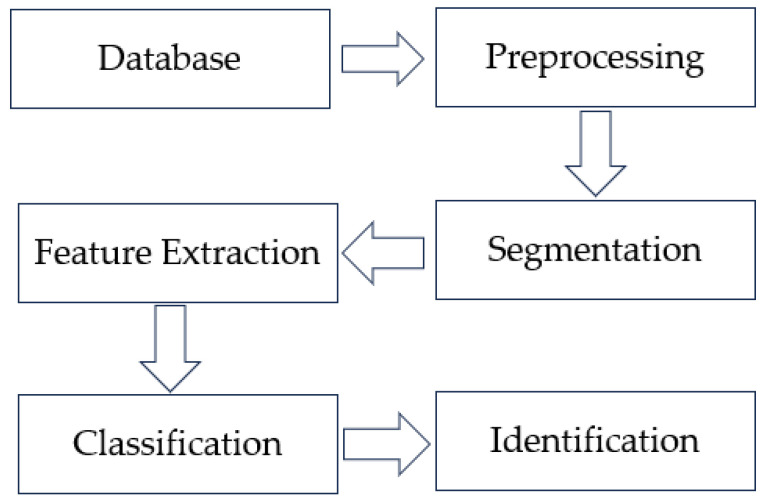
A generalized block diagram of identification in image processing [96].

**Table 1 foods-12-02976-t001:** Principles of indicators and sensors based on food type.

Status	Type	Metabolites	Shelf Life (20–25 °C)	Evaluation Method		Reference
				Indicators	Sensor	
Solid	Vegetable	Oxygen	3–30 days	Optical sensor by fluorescence, colorimeter based on pH	Electrochemical sensor, laser	[30]
Solid	Fruits	Oxygen	2–20 days	Optical sensor by fluorescence, colorimeter based on pH	Electrochemical sensor, laser	[26]
Solid	Food Animals	ATP-associated compound	2–3 days			[28]
		Glucose/ lactic acid		Colorimeter based on pH	Electrochemical sensor by redox reaction	[26]
		Carbon dioxide		Colorimeter based on pH	Electrochemical sensor by silicon-based polymers	[30]
		Biogenic amines		Color-changing pH-sensitive dyes	Electrochemical sensor by enzyme redox reaction	[31]
Liquid	Dairy product	Glucose/ lactic acid	1 day	Colorimeter based on pH	-	[29]
Jelly	Fermented food	Glucose/ lactic acid	7–180 days	Colorimeter based on pH	Electrochemical sensor by redox reaction	[32]
		Carbon dioxide		Colorimeter based on pH	Electrochemical sensor by silicon-based polymers	[26]

## Data Availability

The datasets generated for this study are available on request to the corresponding author.

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
