# Peer review of "Recent Advance of Intelligent Packaging Aided by Artificial Intelligence for Monitoring Food Freshness"

_foods, 2023, doi:10.3390/foods12152976_

Round 1

Reviewer 1 Report

The paper started as an interesting paper to read. The artificial intelligence plays a more and more important role in our modern society. Moreover, intelligent packaging is the future of packaging in the XXI century. Unfortunately, the paper fails to provide useful information, and the feeling is that is just an elaborate introduction and not a review paper. In my opinion, first, the paper should define, clearly the food freshness. As the authors state: “the evaluation of food freshness is an important direction for food quality evaluation in the future. The freshness of food is related to its own metabolism and is influenced by the type of food, microbial species, storage conditions and packaging methods. Freshness determination techniques are usually used to predict the freshness of food by the relationship between the metabolites of the food (e.g. glucose, ethanol, organic acids, adenosine triphosphate degradates, biogenic amines) and the indicators”, a general discussion about food freshness may become irrelevant. As I assume that a comprehensive discussion is not the purpose of the present review, at least the authors should mention some categories of food as they consider (but some general classification has to be discussed since I assume that the packaging is different for solid, liquid or jelly like foods; then a discussion related to the type of food such as meat, fish, vegetables, fruits, etc should be provided; then a discussion related to the short, medium and long shelf life, should be also provided). Then a discussion related to the properties (mechanical, thermal, electrical, optical and so on) of packaging in respect with the chosen category of food should be provided.               

In the present form the paper looks like a collection of facts, especially the first part. I start to like the paper starting to the 3.3.1 Bar code technology. But also here is a general discussion. It would be more attractive, if the authors will explain what king of information is encoded and how, in each of 1D and 2D codes. And would be great if you will not repeat the same information: i) “They can be divided into 1D and 2D barcodes according to their shape” and “Generally speaking, barcodes can be divided into one and two dimensions.”               

RFID is interesting and an image could be important.               

Augmented reality can be interesting and an image and more details about the used can be interesting.              

In general in my opinion each class (subchapter) should be accompanied by at least one figure and to be discussed in terms of food classification, packaging relevant properties, to highlight the factor that make this intelligent, the way in which the information is obtained (visual inspection, need of a simple device, need of complex device, can be performed in situ, can be performed only in laboratory, do we need specialized instruments, spectrometers, etc.), accuracy and limitation (also in relation with type of food).   Additionally, the main purpose of the present review, as it is declared by authors “to provide a concise and comprehensive summary of the latest advancements in artificial intelligence technology for intelligent packaging, with a focus on freshness detection”, but the section dedicated to artificial intelligence had less to do with packaging. In fact the term “packaging” appears only one in the first sentence of section 4.1 (the first and general sentence), 0 times in section 4.2. Then where is the “latest advancements in artificial intelligence technology for intelligent packaging”. But indeed the focus is on freshness detection (mainly by a direct visualization of food).               

Other remarks:               

Please read again the paper, and try to reduce some long sentences, and improve a beat the language.               

Remove the long paragraphs with general remarks and no practical purpose such as: “Intelligent packaging has high research value and application prospects in the packaging industry, especially in the area of food safety, which has great potential. As modern technology and people's living standards continue to improve, people are more concerned about food freshness and other indicators, and intelligent packaging has become a focus for researchers in various countries because of its ability to accurately determine product quality and thus ensure food safety and extend product shelf life.” These kinds of ideas are ok to be expressed one time at the beginning. Keep repeating them makes the paper to be hard to be read.               

Page 4 line 154 clarify what kind of spectroscopy (NMR, FT-IR, UV-VIS)?               

Page 4 line 168 “pork” (meat, organs, bacon?) I assume that will have different freshness property!               

Page 9 line 392 “downloaded from Kaggle” explain what is that?               

Page 10 line 435 “supervised learning algorithms such as artificial neural networks (ANN)”. ANNs is more a programing environment and supervised learning is a training method.

Page 12 line 515: Please explain “chemical signals”!               

For freshness detection, more than a classification, the “regression” method associated to the use of ANN is more appropriate to be used since the degree of freshness can be predicted! Can you comment on this?

Readable, but in places can be revised.

Author Response

Thank you very much for your comments, Please see the attachment for details.

Reviewer 2 Report

The work is quite interesting, but there are errors in the proper preparation of the literature, in many items the names of scientific journals are written with a capital letter, e.g. item 4, 5, 6, etc.

The authors of the paper could provide more examples/research methods, subchapter 3.1.1.1 is particularly short.

A table should be added to it, which will certainly allow readers to quickly interpret the text and provide new information.

The table should include, among others:

- method of assessment/methods used to assess the freshness of products,

- tested/analyzed features/

- references.

Author Response

(The authors gave the same response as above.)

Round 2

Reviewer 1 Report

Thank you for considering my suggestions.

The language is fine.